# The Correlation of Short-Chain Fatty Acids with Peripheral Arterial Disease in Diabetes Mellitus Patients

**DOI:** 10.3390/life12101464

**Published:** 2022-09-20

**Authors:** Akhmadu Muradi, Chyntia Olivia Maurine Jasirwan, Charley D. Simanjuntak, Dedy Pratama, Raden Suhartono, Patrianef Darwis, Aria Kekalih

**Affiliations:** 1Vascular and Endovascular Surgery Division, Medical Staff Group of Surgery, Cipto Mangunkusumo Hospital, Faculty of Medicine, Universitas Indonesia, Depok 16424, Indonesia; 2Hepatobiliary Division, Medical Staff Group of Internal Medicine, Cipto Mangunkusumo Hospital, Faculty of Medicine, Universitas Indonesia, Depok 16424, Indonesia; 3Artificial Intelligence in Medicine & Digital Health, Medical Instrumentation & Simulators Research Group, Department of Community Medicine, Faculty of Medicine, Universitas Indonesia, Depok 16424, Indonesia

**Keywords:** SCFA, diabetes mellitus, peripheral arterial disease

## Abstract

Diabetes mellitus (DM) is a significant risk factor for peripheral arterial disease (PAD). PAD affects 20% of DM patients over 40 and has increased by 29% in the last 50 years. The gut microbiota produces short-chain fatty acids (SCFAs) that affect atherosclerosis. SCFA inhibits inflammation, which contributes to atherosclerosis. This study tried to link feces SCFA levels to atherosclerosis in people with diabetes with peripheral arterial disease (PAD). The study included 53 people with diabetes and PAD: gas chromatography-mass spectrometry measured acetate, butyrate, and propionate levels in feces samples (GC-MS). There was a positive correlation between random blood glucose (RBG) levels, peak systolic velocity (PSV), volume flow (VF), plaque, relative and absolute acetate, relative valerate, butyrate, and propionate. This supports the idea that elevated SCFA levels in type 2 diabetic (T2D) patients reduce adipose tissue inflammation and cholesterol metabolism, contributing to atherosclerosis pathogenesis. We conclude that increased fecal SCFA excretion is linked to cardiovascular disease. To determine the causal effect correlation of the SCFA with clinical and laboratory parameters for PAD in DM patients, compare the SCFA in plasma and feces, and account for confounding variables, a specific method with larger sample sizes and more extended follow-up periods is required.

## 1. Introduction

Diabetes mellitus (DM) is a group of metabolic diseases characterized by a chronic elevation of blood glucose levels above normal limits (hyperglycemia) caused by impaired insulin secretion, insulin resistance, or a combination of the two [1]. Diabetes mellitus is a significant risk factor for a variety of atherosclerotic diseases [2,3]. Peripheral arterial disease (PAD), which results from atherosclerosis in arteries other than the aorta and cerebral arteries, is one of the most distressing complications of diabetes mellitus [4].

Patients with PAD develop various complications, including intermittent claudication or pain in the feet with exertion, ischemic resting pain, ulceration, and gangrene, all requiring revascularization followed by amputation. The diagnosis is made by recalling the risk factors for PAD and the characteristics of the symptoms. Following that, a PAD diagnosis is conducted by inspecting and palpating peripheral pulses on the dorsalis pedis, posterior tibial, popliteal, and femoral arteries [5,6,7]. One diagnostic parameter for PAD is peak systolic velocity (PSV) [8]. However, the PSV value in sclerosis arteries will be lower. PAD is present in approximately 20% of DM patients over the age of 40 and increases to 29% over 50 [9]. Despite their widespread prevalence, strategies for improving nutrition in PAD patients have received scant attention [10].

Inflammation is a risk factor for atherothrombosis, and C-reactive protein (CRP) is a biomarker of inflammation associated with glucose regulation. Similarly, DM impairs fibrinolytic function and induces the production of plasminogen activator inhibitor-1 (PAI-1), increasing the risk of atherosclerotic plaque rupture and thrombus formation [2].

SCFAs are a major class of intestinal microbial metabolites; they are a by-product of dietary fiber fermentation that the gut microbiota cannot digest. SCFA levels were highest in the proximal colon [11]. SCFA inhibits the inflammatory process by preventing immune cells from migrating, proliferating, and producing a variety of cytokines [12]. Acetate, propionate, and butyrate [13] are excreted in feces or absorbed by the intestinal epithelium via selective anion transport proteins [14]. Numerous studies have discovered an inverse relationship between SCFA and the size of atherosclerotic plaques [15]. Butyrate and propionate act as immune cell regulators in atherosclerosis, alleviating inflammation in the diseased endothelial cells [16]. By producing more SCFA, a more diverse microbiome is associated with decreased insulin resistance. SCFAs have been shown to have metabolic effects such as enhancing energy metabolism, activating gluconeogenesis in the duodenum, preventing endotoxemia metabolism, and reducing inflammation via hormone regulation [17].

This study aims to determine the relationship between SCFA levels and PAD in DM patients at Cipto Mangunkusumo Hospital, Indonesia’s national and tertiary referral hospital. To determine the degree of PAD in DM patients, a fecal SCFA examination, ankle pressure, toe pressure, and ankle-brachial index (ABI) were performed.

## 2. Materials and Methods

### 2.1. Samples and Population

A cross-sectional study was conducted on 53 PAD patients with diabetes mellitus at Cipto Mangunkusumo Hospital between September 2020 and October 2021. Diabetes mellitus is defined in diabetic patients as fasting blood glucose levels greater than 126 mg/dL, two-hour postprandial blood glucose levels greater than 200 mg/dL, and an HbA1C level greater than 6.5%. Additionally, peripheral arterial disease is defined as intermittent claudication, resting foot pain, an ankle-brachial index (ABI) of 0.9, an ankle pressure of 60 mmHg with or without ulcer or gangrene, or 40 mmHg without ulcer or gangrene. All DM patients included in this study were on optimal control.

People with sepsis, human immunodeficiency virus (HIV)/acquired immunodeficiency syndrome (AIDS), those who have taken antibiotics or probiotics in the last month, and people with diabetes who are in good health based on their medication and lifestyle are not allowed. Meanwhile, the study enrolled eligible participants from both outpatient and inpatient settings.

This study has been ethically approved through the Health Research Ethics Committee of the Faculty of the Medicine University of Indonesia with the number: KET-974/UN2.F1/ETIK/PPM.00.02/2021/PRO21-10-1063.

### 2.2. Rutherford Classification

The Rutherford classification classifies PAD severity according to the following clinical symptoms: (1) 0–3 scale: asymptomatic, mild, moderate, or severe claudication; (2) 4–6 scale: ischemic resting pain, minor tissue loss [18].

### 2.3. Short-Chain Fatty Acids (SCFA) Analysis


**SCFA Extraction Process from Feces Sample**


All participants did not go through the process of eating or fasting before the samples were collected. Feces samples were collected from all participants, stored in sterile plastic containers, and immediately frozen at 20 °C for further SCFA analysis. After being thawed, the sample was made into a homogeneous solution using a spatula and aliquot, weighed in a vial, and prepared in serial of 6 levels of concentration 8–0.25 μmol/L, then made the standard diluent. After the solution is homogeneous, add 100 μL of supernatant and 300 μL of H_2_O and homogenize, and add 425 μL of isopropanol alcohol (IPA) and 75 μL of 1.5 N HCl. After that, the sample was transferred to a yellow gas chromatography (GC) tube for the sample group and GC-amber for the control group, tightly closed, and 1.2 μL of the solution was injected into the GC-MS system. This step was repeated for the sample and control groups. A total of 1 mL of standard diluent was added to 200 mg of the stool sample, followed by sonification for 20 min, centrifugation at 10,000× *g* for 5 min, and sonification for 20 min. All the sample and control preparation were prepared as per the manufacturer’s instructions and standards in Prodia Laboratories Indonesia.


**SCFA Analysis Process with GC-MS**


SCFA was analyzed using the gas chromatography-mass spectrometry (GC-MS) method. This gas chromatography (GC) tool is equipped with a split/splitless injector and a mass spectrometer detector code 5973 [19]. The capillary chromatography column used is a nitro terephthalic acid (PEG) modified polyethylene glycol column. The injector on this GC device is set at 280 °C. The injection was carried out in splitless mode (3 min undivided time). The temperature of the tools was started at 40 °C for 3 min, programmed at a speed of 20 °C/min to 160 °C, and then at 40 °C/min to 245 °C held for 1.87 min resulting in a total centrifugation of 13 min. The gas medium used in this test is hydrogen at a flow rate of 3.70 mL/min. The identity of SCFA detected in the original sample was quantitatively analyzed by mass spectroscopy based on the comparison of time and mass spectrum of an authentic standard using an internal calibration method. SCFA concentration is calculated by constructing a calibration curve with the area under the analyte curve on the standard analyte at each level.


**Measurement of Lipid and Glucose Profile**


Meanwhile, to measure the lipid profile (LDL, HDL, and triglycerides), the author used the colorimetric enzymatic method, where the cholesterol will be enzymatically hydrolyzed into glycerol, and free acids with special lipases will form color complexes that can be measured using a spectrophotometer [20]. Furthermore, the measurement of the random blood glucose profile used the point of care testing (POCT) method, where the total blood glucose level is measured based on electrochemical detection by coating with glucose oxidase enzyme on membrane strips [21].

### 2.4. Estimation of Sample Measurement

As a result of the calculation of the correlation formula with a coefficient of 0.4, a minimum estimated sample size of 50 patients was determined, and eligible subjects were sequentially enrolled in the sample throughout the study period.

### 2.5. Statistical Analysis

The Statistical Package for the Social Sciences (SPSS) version 28 was used to analyze the data. Normality and validity tests were performed using the Kolmogorov–Smirnov normality test. All data are distributed as a mean with a standard deviation (SD). The median (minimum, maximum) represents data with a skewed distribution. Correlation analyses examined the relationship between SCFA and PSV, VF. For non-parametric correlation, a Spearman rank-order test is used.

## 3. Results

### 3.1. Patient Characteristics

The respondents’ characteristics, including SCFA profiles, are summarized in Table 1. Detailed information about SCFA profiles in each subject also can be found in Appendix A: SCFA profiles, while other pathological parameters can be found in Appendix A: Pathological parameters.

The characteristics of the respondents are described in Table 1, where the majority are men (56.6%) with a mean age of 59.15 ± 10.29 years. In this study, there are no significant sex-specific differences. The median age is 60 years, with the youngest and oldest being 27 and 73 years old, respectively. The majority of respondents, 38 (71.7%), experienced anemia with an average of 11.53 g/dL, which was determined based on hemoglobin levels < 13 g/dL in men and <12 g/dL in women [22]. The median leukocyte is 9.36/µL, with 13 samples having leukocytosis (35.13%), and the average platelet sample is 346.91 × 103/µL with a standard deviation of 138.99 × 103/µL. The lipid profile of the sample consists of an average LDL level of 108.22 mg/dL, an average HDL of 40.65 mg/dL, and a median of 130 mg/dL of triglycerides. The sample’s average random blood glucose (RBG) is 218.35 mg/dL, with a standard deviation of 69.67 mg/dL.

According to the Rutherford classification, there are more patients with severe PAD (scale 4–6) (58.5%) than patients with mild PAD (41.5%). Moreover, the vascular examination of the sample obtained a median ankle pressure of 90 mmHg with a maximum value of 140 mmHg, an average toe pressure of 61.22 mmHg with a standard deviation of 24.67 mmHg, and a median brachial pressure of 120 mmHg with the lowest value 100 mmHg and the highest 140 mmHg. Ankle-brachial index (ABI) samples have a median of 0.75, with the highest value of 1.17.

The types of SCFA observed in this study are acetate, propionate, butyrate, and valerate. Based on the results of descriptive analysis, the relative acetate (%) has an average of 54.21% with a standard deviation of 21.87%, and absolute acetate has a median of 3.49 mg/mL with the lowest value of 0.28 mg/mL and the highest 19.35 mg/mL. The relative propionate level (%) has an average of 17.67% with a standard deviation of 11.46%, and absolute propionate levels had a median of 1.51 mg/mL with a minimum value of 0.04 mg/mL and a maximum of 12.16 mg/mL. Meanwhile, the relative butyrate level (%) has a median of 13% with absolute butyrate and 1.8 mg/mL. Furthermore, the relative valerate (%) has a median of 1.5% and the absolute valerate with a median of 0.3 mg/mL. The total SCFA levels in the samples have a median of 7, where the lowest and highest values are 50 mg/mL.

### 3.2. Correlation of Short-Chain Fatty Acid with Lipid and Glucose Profiles of Diabetes Mellitus Patients with Peripheral Arterial Disease

SCFA correlation with triglycerides and random blood glucose is shown in Table 2.

### 3.3. Correlation of Short-Chain Fatty Acid with Foot Arterial Diameter, Peak Systolic Velocity (PSV), and Volume Flow (VF) in Diabetes Mellitus Patients with Peripheral Arterial Disease

Table 3 shows the correlation between SCFA and peak systolic velocity and volume flow of the common femoral artery (CFA), superficial femoral artery (SFA), popliteal artery (POPA), posterior tibial artery (PTA), and dorsal pedis artery (DPA).

A weak positive correlation between PSV DPA and relative acetate (r = 0.38, *p* = 0.038) and PSV POPA and absolute propionate (r = 0.086, *p* = 0.049) was discovered in this study. Furthermore, a significant and weak negative correlation was discovered between PSV CFA and relative valerate (r =−0.4, *p* = 0.016), as well as between PSV SFA and relative valerate (r = 0.392, *p* = 0.02). However, no correlation was observed between other arterial PSV and SCFA components.

The results indicated that there was a very weak positive correlation between relative acetate and VF SFA (r = 0.09, *p* = 0.01) and VF POPA (r = 0.03, *p* = 0.01). Additionally, a significant and weak positive correlation between relative valerate and VF SFA (r = 0.214, *p* = 0.002) and total SCFA and VF CFA (r = 0.122, *p* = 0.04) was discovered. No significant correlation was observed between the VF of other arteries and SCFA components.

### 3.4. Comparative Analysis of Short-Chain Fatty Acid with Foot Arterial Plaque in Diabetes Mellitus Patients with Peripheral Arterial Disease

The relationship of SCFA with foot arterial plaque, including CFA, SFA, POPA, PTA, and DPA, is shown in Table 4.

Figure 1 shows the significant difference of absolute acetate levels in the group with a plaque on CFA compared to those without plaque. In the group with plaque on CFA, the absolute acetate level was 6.505 (1.04, 16.73) mg/mL, while in those without plaque, the concentration was 2.08 (0.28, 19.35) mg/mL (*p* = 0.039). There was also a significant difference in absolute butyrate levels in the group with plaque compared to the group without plaque (*p* = 0.000) in the SFA vessels. The group with plaque on SFA had absolute acetate levels of 4.5 mg/mL (0.1, 11.2) and those without plaque were 2.17 mg/mL (0.01; 13.6). Similarly, relative valerate levels were also significantly different in SFA vessels with a median of 2.35 mg/mL (0.01, 8.7) (*p* = 0.03), while those without plaque had a median of 1.15 mg/mL (0.7; 5.8).

There was a significant correlation between POPA plaque and absolute butyrate (3.8 (0.1, 13.6) mg/mL vs. 1 (0.01, 7.4) mg/mL, *p* = 0.046) and total SCFA was 17 (1, 46) mg/mL vs. 6 (1, 50) mg/mL, *p* = 0.046. A significant correlation was also discovered between POPA plaques and relative valerate with a median of 2.44 mg/mL (0.01, 6.7), in those with plaque and a median of 1.67 mg/mL (0.01, 4.5) in those without plaque (*p* = 0.03).

## 4. Discussion

This study analyzed 53 subjects with a median age of 60 years who met the inclusion and exclusion criteria. According to the literature, the risk of PAD increases with age in patients with diabetes mellitus [3]. Around 70% of the subjects were male, consistent with the reported prevalence of PAD being higher in males [13,14].

The lipid profiles were more or more consistent with some previous studies, which stated that higher triglycerides are risk factors associated with PAD incidence in Framingham [13,23,24].

This study grouped PAD suffered by the subject into two categories based on the Rutherford classification system. Approximately 46% of the subjects had severe PAD with ischemic pain at rest, tissue loss, chronic ulceration, gangrene, and chronic limb-threatening ischemia (CLTI) conditions that can accompany a history of significant amputation. It was reported that only DM increases the risk of CLTI by 2 to 4 times, where approximately 12–20% of cases of CLTI are secondary to DM [16,25,26], which occurs in 11% of PAD patients.

Since SCFA is a metabolic product of the gut microbiota, the composition and the daily diet determine a person’s SCFA levels [27]. Based on the correlation analysis between SCFA levels and lipid and glucose profiles of DM patients with peripheral arterial disease, there is a very weak positive correlation between RBG and absolute propionate, absolute butyrate, and total SCFA levels. Although the correlation is very weak, these results align with the theory that hyperglycaemic conditions in DM patients increase the risk of atherosclerosis. Meanwhile, a significant negative correlation was discovered with very weak strength between triglycerides and absolute valerate levels. This is in line with Zwartjes [28] and Vourakis [29], where a decrease in triglyceride levels occurs when SCFA in plasma and feces are high. A valerate is a form of SCFA that inhibits cholesterol transport, adipocyte dysfunction, and adipose tissue inflammation. It was also discovered that SCFA produced by gut microbiota could influence cholesterol metabolism [30,31].

Based on the peak systolic velocity (PSV), the analysis showed a significant positive correlation between the relative acetate value and PSV PTA, absolute propionate and PSV POPA, valerate relative to PSV CFA and PSF SFA. However, there was no significant correlation between SCFA and other arterial PSV. Therefore, this study relates to the correlation of SCFA to CFA and DPA diameters, which are in line with de La Cuesta-Zuluaga [32], Calderón-Pérez [33] and Huart [34]. SCFA produced by gut microbes reduce the inflammation process of atherosclerosis, thus, lowering the peak systolic velocity [35].

The results showed a significant difference in absolute acetate levels for those with CFA plaque; also, absolute butyrate, relative valerate, and total SCFA with SFA and POPA plaques. This is in line with previous results by de La Cuesta-Zuluaga [32] Calderón-Pérez [33] and Huart [34]. SCFA contributed the plaque presence in vascular, in which the microbiota-produced SCFAs act as mediators of inflammation at certain receptors that may be plaques such as the Takeda bile acid receptor G-protein-coupled receptor 5 (TGR5) and farnesoid X receptor (FXR), trimethylamine-N-oxide (TMAO). and trimethylamine (TMA) through a direct permeability process from the gut. The resulting short-chain fatty acid production can increase the inflammatory effect on these receptors [35].

There is a very weak positive correlation between relative acetate levels with VF SFA and POPA, which is in line with the study of Shi Yunfan [36] and Lin Li et al. [37]. Relative acetate inhibits the production of oxalate to cause atherosclerosis. Acetate also increases lipid oxidation, which reduces the risk of fat accumulation in peripheral arteries [22,38]. A weak positive correlation was also discovered between peak systolic velocity (PSV) of PTA and POPA with absolute propionate; PSV CFA and SFA with relative valerate; volume flow (VF) SFA and POPA with relative acetate; VF SFA with relative valerate; and VF CFA with total SCFA.

## 5. Conclusions

The results showed a significant positive correlation between random blood glucose levels with absolute propionate, absolute butyrate, and total SCFA. However, a significant negative correlation occurred between triglycerides and absolute valerate. A significant positive correlation was also discovered between the diameter of SFA and DPA with relative acetate. Meanwhile, a significant positive correlation was found in the relative acetate value with peak systolic velocity (PSV) PTA, PSV POPA with absolute propionate, PSV CFA, and PSV SFA relative valerate. Relative acetate and valerate levels have a weak positive correlation with VF SFA. The relative acetate levels were also weakly correlated with VF POPA, while total SCFA correlated with VF CFA.

The group with a plaque on CFA had a higher absolute acetate than the group without plaque CFA. Furthermore, the group with POPA and SFA plaques had higher absolute butyrate and relative valerate than those without POPA and SFA plaques. The group with POPA plaques also had a higher total SCFA value than the group without POPA plaques. There was a correlation between total SCFA with SFA and DPA spectral waves, absolute propionate with PTA spectral waves, absolute butyrate with PTA spectral waves. There was also a significant correlation between absolute acetate in the DPA spectral wave. This is aligned with previous study that said SCFA levels in DM patients inhibits the inflammation process of adipose tissue and influence the cholesterol metabolism, which can increase the risk of atherosclerosis. Meanwhile, there was a negative correlation between the diameter of CFA and absolute acetate and the DPA diameter and relative propionate. It is also aligned with the previous studies that mentioned the more SCFA excreted in feces associated with higher risk of cardiovascular diseases.

Further study with a larger sample size, a more extended period, and a case-control or cohort design involving healthy subjects is recommended to compare SCFA values in healthy people and DM patients. There is also a need to evaluate the causal influence of the correlation of SCFA with clinical and ultrasonographic parameters for PAD in DM patients. When measuring SCFA, it is necessary to compare the examination between SCFA in plasma and feces to accurately characterize the rate of excretion, production, and absorption of SCFA in the intestine. Since PAD is a multifactorial disease, it is also essential to consider better confounding variables.

## Figures and Tables

**Figure 1 life-12-01464-f001:**
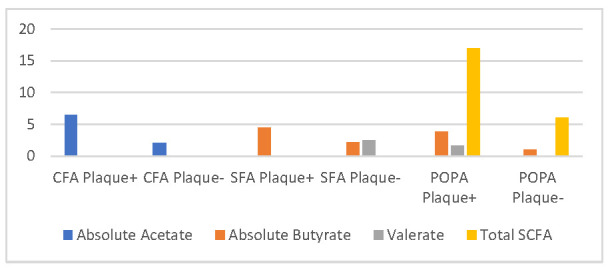
The comparison of scfa with plaque (significant).

**Table 1 life-12-01464-t001:** Characteristics of respondents and SCFA profiles.

Characteristics	Mean/N	%
Age (year)	59.15 ± 10.29	
Sex		
Men	30	56.6
Women	23	43.4
Sore Feet		
Right	21	39.6
Left	32	60.4
LDL (mg/dL) *	108 (35–183)	
HDL (mg/dL) *	39 (5–83)	
Triglycerides (mg/dL) *	130 (60–361)	
RBG (mg/dL)	228.77 ± 73.11	
Ankle Pressure (mmHg) *	90 (0–140)	
Toe Pressure (mmHg) *	60 (30–130)	
Brachial Pressure (mmHg) *	120 (100–160)	
ABI *	0.75 (0–1.17)	
PAD Degree		
Rutherford 0–3	22	41.5
Rutherford 4–6	31	58.5
Short-Chain Fatty Acids		
Acetate (%)	57.45 ± 19.69	
Absolute Acetate (mg/mL) *	3.08 (0.28–19.35)	
Propionate (%)	16.83 ± 10.04	
Absolute Propionate (mg/mL) *	1.25 (0.04–12.16)	
Butyrate (%)	14.28 ± 10.13	
Absolute Butyrate (mg/mL) *	1 (0.01–13.6)	
Valerate (%) *	2 (0.01–9.6)	
Absolute Valerate (mg/mL) *	0.23 (0.01–5.03)	
Total SCFA (mg/mL) *	7 (1–50)	

LDL, low-density lipoprotein; HDL, high-density lipoprotein; RPG, random plasma glucose; ABI, ankle-brachial index; PAD, peripheral arterial disease; SCFA, short-chain fatty acids. Data are displayed in mean ± SD. * Data are displayed in median (Min; Max).

**Table 2 life-12-01464-t002:** Correlation of short-chain fatty acid with lipid and glucose profiles in diabetes mellitus patients with peripheral arterial diseases.

SCFA	Triglycerides		RBG	
	r	*p*	r	*p*
Acetate (%)	0.03	0.29	0.043	0.22
Absolut Acetate (mg/mL) *	0.05	0.21	0.06	0.15
Propionate (%)	0.01	0.54	0.06	0.13
Absolut Propionate (mg/mL) *	0.07	0.104	0.12	0.04
Butyrate (%)	0.08	0.09	0.01	0.55
Absolute Butyrate (mg/mL) *	0.07	0.11	0.13	0.03
Valerate (%) *	0.02	0.37	0.05	0.17
Absolute Valerate (mg/mL) *	0.12	0.04	0.06	0.16
Total SCFA (mg/mL) *	−0.101	0.044	−0.115	0.026

Based on the results of the analysis, it was discovered that there is a very weak positive correlation between blood glucose levels and absolute propionate (r = 0.12, *p* = 0.04), absolute butyrate (r = 0.13, *p* = 0.03), and total SCFA (r = 0.115, *p* = 0.026). There is also a very weak negative correlation with very weak strength between triglycerides with absolute valerate (r = −0.12, *p* = 0.04) and total SCFA (r = −0.101, *p* = 0.044). * Pearson correlation analysis.

**Table 3 life-12-01464-t003:** Correlation of short-chain fatty acid with foot arterial diameter, peak systolic velocity and volume flow in diabetes mellitus patients with peripheral arterial disease.

Artery	SCFA																	
	Acetate (%)		Absolut Acetate (mg/mL)		Propionate (%)		Absolut Propionate (mg/mL)		Butyrate (%)		Absolut Butyrate (mg/mL)		Valerate (%)		Absolut Valerat (mg/mL)		Total SCFA	
	r	*p*	r	*p*	r	*p*	r	*p*	r	*p*	r	*p*	r	*p*	r	*p*	r	*p*
CFA *																		
PSV	−0.22	0.188	−0.4	0.016	0.273	0.107	−0.113	0.51	−0.016	0.928	−0.19	0.278	0.193	0.259	−0.018	0.92	0.002	0.656
VF	−0.001	0.994	−0.165	0.336	−0.118	0.493	−0.293	0.083	−0.327	0.051	−0.327	0.05	0.202	0.238	−0.092	0.595	0.122	0.04
SFA																		
PSV	0.118	0.03	0.113	0.51	0.125	0.467	0.206	0.229	0.11	0.52	0.125	0.469	−0.02	0.92	0.194	0.256	0.065	0.675
VF	0.09	0.01	0.238	0.162	−0.05	0.774	−0.112	0.514	−0.192	0.26	−0.19	0.26	0.214	0.002	0.025	0.886	−0.21	0.215
POPA																		
PSV	−0.185	0.31	0.089	0.626	0.105	0.566	0.123	0.086	0.349	0.05	0.281	0.119	−0.11	0.565	0.207	0.255	−0.101	0.04
VF	0.03	0.01	0.159	0.384	−0.022	0.906	−0.07	0.705	−0.115	0.53	−0.12	0.53	0.076	0.679	0.081	0.66	−0.13	0.466
PTA																		
PSV	0.143	0.45	−0.01	0.94	−0.07	0.73	0.049	0.591	−0.21	0.275	−0.148	0.435	−0.05	0.803	−0.14	0.467	−0.115	0.03
VF	−0.296	0.112	0.26	0.166	−0.007	0.97	−0.014	0.943	−0.056	0.769	−0.06	0.769	0.156	0.411	0.66	0.01	−0.07	0.7
DPA																		
PSV	0.381	0.038	0.126	0.507	−0.376	0.04	−0.086	0.652	−0.114	0.55	−0.06	0.767	−0.14	0.457	−0.08	0.67	-	-
VF	−0.199	0.291	0.17	0.368	0.269	0.15	0.018	0.924	0.943	0.018	0.769	0.153	0.016	0.932	0.957	0.303	0.224	0.234

Spearman correlation analysis. * Pearson correlation analysis. SCFA, short-chain fatty acid; CFA, common femoral artery; SFA, superficial femoral artery; POPA, popliteal artery; PTA, posterior tibial artery; DPA, dorsalis pedis artery.

**Table 4 life-12-01464-t004:** Comparative analysis of short-chain fatty acid with plaque in diabetes mellitus patients with peripheral arterial disease.

SCFA	CFA			SFA			POPA			PTA			DPA		
	Plaque+ (n = 16)	Plaque- (n = 20)	*p*	Plaque+ (n = 11)	Plaque- (n = 25)	*p*	Plaque+ (n = 11)	Plaque- (n = 21)	*p*	Plaque+ (n = 17)	Plaque- (n = 13)	*p*	Plaque+ (n = 15)	Plaque- (n = 15)	*p*
Acetate (%)	58 ± 13.92	53.45 ± 25.16	0.497	62.4 ± 12.45	52.8 ± 22.85	0.22	60.63 ± 16.28	51.85 ± 23.99	0.286	55.47 ± 21.65	57.38 ± 20.44	0.808	55.2 ± 22.96	57.4 ± 19.13	0.778
Absolute acetate (mg/mL) *	6.505 (1.04,16.73)	2.08 (0.28,19.35)	0.028	3.27 (1.04,16.73)	4.455 (0.28,19.35)	0.52	8.5 (0.48,15.73)	1.86 (0.28,19.35)	0.16	5.43 (0.48,15.73)	2.5 (0.28,19.35)	0.385	3.5 (0.48,15.73)	3.49 (0.28,19.35)	0.967
Propionate (%)	18.43 ± 9.99	17.9 ± 12.45	0.889	16.9 ± 9.99	18.61 ± 11.87	0.689	15.09 ± 8.82	18.47 ± 11.66	0.406	17.94 ± 10.22	16.92 ± 12.14	0.805	18.67 ± 10.42	16.33 ± 11.61	0.567
Absolute propionate (mg/mL) *	1.74 (0.1,12.16)	1.17 (0.04,7.82)	0.132	1.74 (0.1,5.6)	1.455 (0.04,12.16)	0.931	1.83 (0.1,12.16)	1.37 (0.04,7.82)	0.254	1.6 (0.1,12.16)	1.37 (0.04,7.82)	0.341	1.6 (0.1,12.16)	1.4 (0.04,7.82)	0.683
Butyrate (%)	18 (2.3)	10.5 (0.49)	0.2	11 (2.28)	13.5 (0.49)	0.958	21 (2.29)	11 (0.49)	0.223	19 (0.33)	10 (1.28)	0.145	18 (0.33)	11 (1.29)	0.595
Absolute Butyrate (mg/mL) *	2.5 (0.1,13.6)	0.9 (0.01,7.4)	0.124	4.5 (0.1,11.2)	2.17 (0.01,13.6)	0.000	3.8 (0.1,13.6)	1 (0.01,7.4)	0.046	2.4 (0.01,13.6)	0.8 (0.01,9.6)	0.263	2.1 (0.01,13.6)	1 (0.01,9.6)	0.902
Valerate (%) *	1.35 (0.2,4.8)	1.55 (0.01,6.7)	0.741	2.35 (0.01,8.7)	1.15 (0.7,5.8)	0.033	2.44 (0.01,6.7)	1.67 (0.01,4.5)	0.03	2.4 (0.01,6.4)	1.4 (0.01,6.7)	0.711	2.4 (0.01,6.4)	1.4 (0.01,6.7)	0.775
Absolute valerate (mg/mL) *	0.28 (0.01,1.9)	0.22 (0.01,5.03)	0.962	0.1 (0.01,1.9)	0.325 (0.01,5.03)	0.59	0.3 (0.01,1.9)	0.3 (0.01,5.03)	0.611	0.3 (0.01,1.9)	0.09 (0.01,5.03)	0.341	0.3 (0.01,1.57)	0.09 (0.01,5.03)	0.624
Total SCFA (mg/mL) *	13.5 (1.46)	6.5 (1.50)	0.116	7 (1.29)	10 (1.50)	0.903	17 (1.46)	6 (1.50)	0.046	13 (1.46)	6 (1.50)	0.3	11 (1.46)	6 (1.50)	0.935

Data are displayed in the form of mean ± SD. * Data are displayed in the form of median (Min; Max). SCFA, short-chain fatty acid; CFA, common femoral artery; SFA, superficial femoral artery; POPA, popliteal artery; PTA, posterior tibial artery; DPA, dorsalis pedis artery.

## Data Availability

Not applicable.

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
