# Peer review of "The Correlation of Short-Chain Fatty Acids with Peripheral Arterial Disease in Diabetes Mellitus Patients"

_life, 2022, doi:10.3390/life12101464_

Round 1
Reviewer 1 Report (Previous Reviewer 3)
The authors have adequately addressed all of the comments raised and in my opinion have significantly improved the manuscript. I think that the manuscript now merits publication in life.
Author Response
Dear Reviewer,
Thank your very much for the kind review and opinion. Here we attach the final version of our manuscript.
Thank You
Best Regards
Akhmadu

Reviewer 2 Report (Previous Reviewer 2)
The authors of this study present relevant findings in the understanding of diabetic complications associated with PAD and how these pathologies alter the levels of short-chain fatty acids. In general, I consider that the manuscript is well structured, but I still have doubts about the methodology used to quantify SCFA because they use very large volumes (liters) for sample processing and nothing is specified about the column used or technical details of gas chromatography coupled to mass and a reference does not appear. Therefore, I consider that the authors should clarify if they used liters of the substances or if there is an error in the units and possibly they used microliters.
I recommend to the authors the following analytical paper to determine SCFA that details the gas chromatography methodology and in this way to be able to make the corrections of the materials and methods section of your manuscript.
Scortichini, S., Boarelli, M. C., Silvi, S., & Fiorini, D. (2020). Development and validation of a GC-FID method for the analysis of short chain fatty acids in rat and human faeces and in fermentation fluids. Journal of Chromatography B, 1143, 121972.
The authors also do not mention the methodology used to measure blood biochemistry and this information must be attached to the manuscript.
Lastly, there are some misspellings in the text and tables that I recommend reviewing.
Once the authors make the suggested changes, the manuscript can be accepted for publication in the journal.
I recommend that the authors attach the bioethics committee's approval information and detail the materials and methods section to facilitate the reproducibility of their methodologies.
Author Response
Dear Reviewer,
Thank you for the review and detailed corrections for improving this manuscript. Here we try to respond point by point.
The authors of this study present relevant findings in the understanding of diabetic complications associated with PAD and how these pathologies alter the levels of short-chain fatty acids. In general, I consider that the manuscript is well structured, but I still have doubts about the methodology used to quantify SCFA because they use very large volumes (liters) for sample processing, and nothing is specified about the column used or technical details of gas chromatography coupled to mass and a reference do not appear. Therefore, I consider that the authors should clarify if they used liters of the substances or if there is an error in the units and possibly they used microliters.
I recommend to the authors the following analytical paper to determine SCFA that details the gas chromatography methodology and in this way to be able to make the corrections of the materials and methods section of your manuscript.
Scortichini, S., Boarelli, M. C., Silvi, S., & Fiorini, D. (2020). Development and validation of a GC-FID method for the analysis of short chain fatty acids in rat and human faeces and in fermentation fluids. Journal of Chromatography B, 1143, 121972.
The authors also do not mention the methodology used to measure blood biochemistry and this information must be attached to the manuscript.
Lastly, there are some misspellings in the text and tables that I recommend reviewing.
Once the authors make the suggested changes, the manuscript can be accepted for publication in the journal.
I recommend that the authors attach the bioethics committee's approval information and detail the materials and methods section to facilitate the reproducibility of their methodologies.
We have revised the unit error and the method in the manuscript and added the methodology of blood biochemistries.
Here we attach the revised manuscript.
Thank you
Sincerely yours
Akhmadu Muradi

This manuscript is a resubmission of an earlier submission. The following is a list of the peer review reports and author responses from that submission.
Round 1
Reviewer 1 Report
Based on the tittle, Muradim, Jasirwan and colleagues have prepared a manuscript aiming to establish the correlation between SCFA levels with PAD, in a sample of patients with DM. Such correlation would only be possible if a group of DM without PAD would be included. The authors, by contrast, in some moment have indeed correlated SCFA levels to PAD degree. However, the manuscript is not in conditions to be published and needs deep improvements.
Starting with the Abstract, an introduction to the subject and importance is absent, as the authors start almost directly with pointing methodology, and then have expanded too much the list of results, without any clear conclusion at the end.
The Introduction is very long, and widely dispersed in the subjects, meaning there is hardly a conducting wire to help the readers follow the idea and relevance. However, the association between gut microbiota and DM should be further introduced, as the sample is exclusively composed of individuals with DM.
The goal of the study should be better justified, and an hypothesis included before establishing the objectives. It is not clear why information about the Doopler is mixed with the expected results. Nevertheless, with such study, it is not possible to establish (or “provide an overview of”) the effects of SCFA on PAD, as there is no intervention. Only associations can be made.
Sentence for defining patients with DM has wrong information (values should be below those levels, not at those levels). Were diabetic patients with an eventually optimal control, based on medication and lifestyle, excluded?
The methods are relatively complete, and only a few concerns here below.
When fecal samples are collected from patients, aren’t patients set up previously in a standard diet? In fact, the authors later state that importance in lines 524-5 (reference 60). It is also not stated in methods, whether patients are randomly fed or fasted (though the reader can later interpret that based on the R from RBG being from random).
A sampling measure estimation was added, which was very valuable. However, a minimum of 50 patients was established, and although it seems 53 individuals were included (based on the information at the methods 2.1), only at the Discussion (line 470) we find out only 37 patients were analysed.
The socio-demographic data and biochemical analyses were not described in methods. It should be clear why the authors are interested in the analysis of urea, creatinine, etc. It is important to note that both triglycerides and glycemia are highly variable based on the fed-fasting state of the individual, and this was never clearly disclosed nor discussed. I would also suggest including Hb1Ac % (no need to fast, and an estimated glycemia or glycemic control can be inferred), and body weight. Nevertheless, this merely descriptive information could be at supplementary data.
The calculation for relative values of each SCFA (expressed as %) should be explained somewhere. The relevance for correlating both relative and absolute should be clearly stated.
The Results section is overall very poor. It consists of a crude description of the overwhelming 17 tables included. The rationale for such analysis, the conclusion of each part, and the connection to the following analysis is mandatory, which is absolutely missing in this manuscript.
For those analysis where comparisons are performed, the title of the tables can’t be “correlation”. For such analysis, I would kindly suggest to the authors to prepare graphs, preferentially showing the individual patients, instead of tables.
Regarding the results interpretation, the authors need to be very careful when find a p value below 0.05, but the r is very low (most times below 0.2), which means there is no correlation as it is so far away from an r of 1.
The Discussion is also poor, and mainly disconnected from each paragraph. Once more, there is a lack of a connection wire to guide the reader.
Minor points:
When talking about prevalence (line 77), a more recent resource should be used, as the data is from 2003.
When referring “insulin action disorders” (line 42), it should be better explained. Reference [1] should be updated to the 2022 supplement of Diabetes Care which is annually revised.
At the Abstract, it is stated “resting blood glucose” as RBG, which the first time I read this; however, in Table 1, RBG is “random blood glucose”, which makes more sense.
A reference might be missing in lines 108 (after “mellitus”), 170 (after “obesity”), 185 (after “effects”).
The meaning if IPA (line 242) is missing.
More complete legends are needed, including the disclosure of the n for each correlation.
At bibliography, the authors are missing in references 1 and 10. Please check date on reference 7. Reference 43 is the same as 51, and neither is correctly referenced. Please check unnecessary information at reference 73.
Reviewer 2 Report
The introduction is well written and clearly shows the background of the study. However, it is very long and it is recommended to reduce it a bit.
The section on materials and methods should be improved, especially in the section on gas chromatography for the determination of short-chain fatty acids, since they mention that they needed liters of all the reagents to process the samples. Therefore, it is necessary that the authors clarify whether it was a unit error and do not mention the specific characteristics of the CG-MS equipment used. Authors are recommended to improve the section and mention the source of the standards used.
In the results section, the authors clearly show the results and I consider it a good idea to highlight the most relevant data in each section in color. However, I find that it is a bit repetitive since the table and the text present the same information. Therefore, the number of tables could be reduced or less relevant data could be annexed as supplementary results.
I agree with the authors' conclusions and it is important to expand the study to strengthen the relevance of the research.
Reviewer 3 Report
In this interesting paper,
Muradi et al. carried out a cross-sectional study with a total sample of 53 patients diagnosed with Diabetes Mellitus and PAD. They analyzed the potential correlation between fecal Short Chain Fatty Acids (SCFA) levels and the rate of atherosclerosis, as reflected by ultrasonic wave parameters in this patients. The results showed that there was a positive correlation between resting blood glucose (RBG) levels and other relevant clinical parameters. The authors found this to be of interest and conclude from their results that further studies should employ a larger sample size and a a more extended period with a case-control/cohort design, to further evaluate any causal effect correlation of the SCFA with clinical and ultrasonography parameters for PAD in Diabetes Mellitus (DM) patients, which would lead to a better understand of the role SCFA in DM.
The experimental approaches are sound and state of the art. In their results and discussion sections the authors delineate the importance of monitoring SCFA in DM patients, which could beneficial for this patients. However, there are considerable flaws in the experimental design and a number of major issues that should be addressed, as outlined in detail below.
Major comment:
- The introduction part is extremely long, which should be focused on the relevant part to this study.
- Ethics number should be included to this paper.
- The storage of the feces samples should be described more in detail? Were the samples immediately measured or stored for longer time under which conditions?
- In line 275 70.27% of the patients are men but in table 1 56.6% are men which one is right?
- If the distribution of men and women is nearly equal in table 1 do they see and gender differences in their analysis?
- The authors should show the most significant results in bar graphs, which would be easier to follow for the readers.